# Stretchable Sensors: Novel Human Motion Monitoring Wearables

**DOI:** 10.3390/nano13162375

**Published:** 2023-08-19

**Authors:** Chia-Jung Cho, Ping-Yu Chung, Ying-Wen Tsai, Yu-Tong Yang, Shih-Yu Lin, Pin-Shu Huang

**Affiliations:** Institute of Biotechnology and Chemical Engineering, I-Shou University, Kaohsiung 84001, Taiwanyangyutong4@gmail.com (Y.-T.Y.); sherry91630@gmail.com (S.-Y.L.);

**Keywords:** wearable sensors, nanotechnology-enhanced strategy, stretchable metal-organic polymer, nanocomposites, reduced graphene oxide (rGO), human body monitoring system

## Abstract

A human body monitoring system remains a significant focus, and to address the challenges in wearable sensors, a nanotechnology-enhanced strategy is proposed for designing stretchable metal-organic polymer nanocomposites. The nanocomposite comprises reduced graphene oxide (rGO) and in-situ generated silver nanoparticles (AgNPs) within elastic electrospun polystyrene-butadiene-polystyrene (SBS) fibers. The resulting Sandwich Structure Piezoresistive Woven Nanofabric (SSPWN) is a tactile-sensitive wearable sensor with remarkable performance. It exhibits a rapid response time (less than three milliseconds) and high reproducible stability over 5500 cycles. The nanocomposite also demonstrates exceptional thermal stability due to effective connections between rGO and AgNPs, making it suitable for wearable electronic applications. Furthermore, the SSPWN is successfully applied to human motion monitoring, including various areas of the hand and RGB sensing shoes for foot motion monitoring. This nanotechnology-enhanced strategy shows promising potential for intelligent healthcare, health monitoring, gait detection, and analysis, offering exciting prospects for future wearable electronic products.

## 1. Introduction

The fields of intelligent healthcare and health monitoring have become prominent research areas. Medical monitoring systems require vital characteristics such as mechanical flexibility, stretchability, high sensitivity, and rapid response [1,2,3,4,5]. These features have garnered significant attention in disease diagnosis, elderly care, health monitoring, and intelligent robotics [6,7,8].

Scientists have developed stretchable metal-organic composite materials with flexibility and customizable structures to meet the demands of the next generation of innovative wearable applications. These materials find widespread applications in intelligent optoelectronics, sensing devices, energy conversion and generation, catalysis, and the biomedical field. However, currently prevalent wearable touch or pressure sensors mainly fall into categories such as piezoelectric, capacitive, triboelectric, inductive, and resistive types, each with advantages and relying on force variations for readout [9,10,11]. Among these, resistive sensors have gained considerable attention due to their simple readout mechanism, ease of manufacturing, low energy consumption, and sensitivity to pressure and bending, effectively transforming mechanical strain into impedance changes.

Sensitive sensing and fast capabilities are crucial for practical medical, disease control, and environmental protection applications. Additionally, introducing various mechanical stimuli [12,13], such as tensile strain, pressure, and bending, enables the recording of object manipulation and body movements. Stretchability [14,15,16] is crucial, requiring good adhesion to electronic devices and the ability to withstand mechanical stress under bending and twisting conditions. In pressure sensor design, introducing two-dimensional polymeric surface microstructures has been a primary method in recent years to enhance the sensitivity of tactile sensors. Techniques such as micro pyramids, micro-domes arrays, micro-groove shapes, and interlocking microstructures have been used [17,18,19]. Despite significant advancements in device sensitivity, conventional manufacturing methods often involve complex and environmentally unfriendly processes, posing challenges regarding environmental friendliness, lightweight, large-scale implementation, and compatibility with monitoring subtle variations.

Fiber-based devices, due to their characteristics such as breathability, durability, sustainability, flexibility, and lightweight, are widely used in wearable electronic products, artificial electronic skin, biomedical components, and intelligent robotics [20,21,22]. Recently, fiber-based sensors utilizing piezoelectric, capacitive, and resistive conversion methods have been extensively developed. However, most fiber-based devices may experience performance degradation during repeated large mechanical movements. Therefore, inherently stretchable electronic products with durability, comfort, small-scale sensing, and robust mechanical performance are crucial for achieving long-term stability, particularly after undergoing long-term tensile deformation in fiber-based electronic devices.

We suggest a novel approach that utilizes nanotechnology to enhance the design of stretchable metal-organic polymer nanocomposites for wearable sensors [23,24,25]. In this method, we harness the properties of reduced graphene oxide (rGO) [26,27,28] to enhance various aspects of the material. Recently, rGO-modified high-performance polymer composites [29,30,31] have developed rapidly. The performance of the composites depends on several factors, such as filler content, interface bonding, and compatibility between the filler and matrix. In addition to their excellent performance, rGO nanofillers possess reactive functional groups, such as epoxy, hydroxyl, and carboxyl [32,33,34]. These polar groups can enhance the compatibility and interface bonding between rGO and silver nanoparticles (AgNPs) in the polymer through chemical or physical methods, providing more possibilities for the composites. Using a porous substrate based on elastic electrospun polymer fibers for manufacturing stretchable composites [35,36,37] has attractive features, including controllable diameter, high surface area-to-volume ratio, and adjustable porosity. Fiber-based electronic products demonstrate excellent long-term stability, withstanding mechanical deformation from pressure to stretching and bending while improving degradation performance. Furthermore, electrospun polymer fiber-based composites show promise in applications such as dressings, filtration-based separation, and fiber-based composite manufacturing [38,39,40,41,42]. These materials have outstanding mechanical properties, high mechanical stability, elasticity, and enhanced electrical and rheological performance, indicating potential applications in electronic sensor systems.

Our research reports a Sandwich Structure Piezoresistive Woven Nanofabric (SSPWN) primarily used for tactile-sensitive wearable sensor technology to improve mechanical performance. The sensor is based on a stretchable metal-organic nanocomposite comprising rGO and in-situ generated silver nanoparticles (AgNPs) [43,44,45,46]. During preparation, we blend rGO with elastic electrospun polystyrene-butadiene-polystyrene (SBS) fibers, improving and enhancing the molecular structure through polymer shear and collision forces. Subsequently, a conductive layer is produced using the in-situ generation method of AgNPs and by controlling the dielectric layer’s spray time to adjust the sensing element’s response level. Finally, the sensor is assembled into a tactile-sensitive wearable sensor with elasticity and mechanical robustness. The sensor has been successfully applied to the fingers, back of the hand, wrist, elbow, and RGB sensing shoes. The tactile-sensitive wearable sensor exhibits outstanding performance, with a fast response time (less than three milliseconds) and highly reproducible stability in 5500 cycles. Additionally, when subjected to a pressure of 0.17 kilopascals, the sensor demonstrates excellent On-Off performance.

This nanotechnology-enhanced strategy aims to improve material performance and has successfully achieved a tactile-sensitive wearable sensor with superior capabilities. This technology holds significant potential in intelligent healthcare and health monitoring, particularly for gait detection and analysis [47,48,49]. With continuous research and improvement, this nanotechnology-enhanced strategy will bring more exciting breakthroughs in developing future wearable electronic products.

## 2. Materials and Methods

### 2.1. Materials and Procedure

We have purchased several materials from Sigma Aldrich for our research, including a linear triblock copolymer, polystyrene-block-polybutadiene. The styrene content in this copolymer is 30 wt%, with an average molecular weight of approximately 140,000, as determined by Gel Permeation Chromatography (GPC). Additionally, it contains less than 0.5 wt% of an antioxidant. Apart from SBS, we have also acquired reduced graphene oxide (rGO), anhydrous tetrahydrofuran (THF, purity 99.9%), N, N-Dimethylformamide (DMF, purity 99.5%), silver trifluoroacetate (AgCF_3_COO, purity 98%), and hydrazine sulfate (NH_2_NH_2_·H_2_SO_4_, purity at least 99.0%). These materials will be used in our experiments and research for developing stretchable metal-organic polymer nanocomposites suitable for wearable sensors.

### 2.2. Preparation of Dielectric Layer by Electrospun SBS Nanofiber Membrane

SBS was dissolved in a mixture of THF and DMF (at a ratio of 3:1) to prepare a 15 wt% SBS solution. The solution was then placed on a heating plate at 160 rpm and 60 degrees Celsius for 8 h. Subsequently, the electrospinning technique was employed to create SBS nanofibrous dielectric layers. The dissolved SBS solution was injected into a metal needle using an infusion pump (KD Scientific Model 100, Holliston, MA, USA) at 0.5 to 0.8 milliliters per minute. The tip of the metal needle was connected to a high-voltage power supply (Chargemaster CH30P SIMCO, Santa Clara, CA, USA), and the voltage during the electrospinning process was set at 13.0 to 15.0 kV. Finally, an aluminum foil was placed 15 cm below the needle tip to collect the SBS nanofibrous dielectric layer produced at different time intervals (1, 2, 3, 4, 5, and 6 min, respectively), as shown in Figure 1a.

### 2.3. Preparation of SBS/rGO Nanofiber Membrane by Electrospinning

Firstly, (1) in the preparation of the solution, SBS and THF were dissolved together to form solution A. Simultaneously, we mixed reduced graphene oxide (rGO) dispersed in DMF (weight percentage of 0.7 wt%) to form solution B. Subsequently, solution A and solution B were mixed after undergoing ultrasonic vibration, resulting in a mixed solution containing 15 wt% SBS (dissolved and dispersed in a solvent mixture of THF:DMF at a ratio of 3:1). The mixed solution was vigorously stirred and then placed on a heating plate at 60 degrees Celsius and 160 rpm for 8 h to ensure complete dissolution and homogenization.

(2) Electrospinning technique was employed to fabricate SBS-rGO nanofiber thin films: The homogenized SBS-rGO solution was injected into a metal needle using an infusion pump (KD Scientific Model 100, USA) at a rate of 0.35 to 0.5 milliliters per minute. The tip of the metal needle was connected to a high-voltage power supply (Chargemaster CH30P SIMCO, USA), and the voltage during the electrospinning process was set at 13.0 to 15.0 kV. Finally, an aluminum foil was placed 15 cm below the needle tip to collect the densely formed SBS-rGO nanofiber thin film for 30 min, resulting in the SBS/rGO Nanofiber membrane.

### 2.4. Preparation of Conductivity Layer by Silver Reduction

An ethanol solution of silver nanoparticles precursor with 15 wt% AgCF_3_COO was prepared, and the SBS/rGO fiber membrane was immersed in the precursor solution. The membrane was dipped in the solution until it became semi-transparent and then left to air-dry naturally for one day. Subsequently, a solution containing 50% Hydrazine sulfate salt mixed with an equal amount of ethanol/water was dropped onto the nanofiber membrane, ensuring complete immersion in the Hydrazine sulfate salt reduction solution for five minutes. The membrane was then rinsed multiple times with deionized water to remove any remaining reducing agent. Finally, the nanofiber membrane (conductive layer), now termed SBS/rGO-AgNPs fiber, was dried using lint-free paper, resulting in a material possessing both elasticity and conductivity characteristics (as shown in Figure 1b).

### 2.5. Characterization

The morphologies of the block copolymer composite ES nanofibers were examined using both SEM (Scanning Electron Microscope—Hitachi S-520, Tokyo, Japan) and an Optical microscope (MA-tek, Taipei, Taiwan). Before image characterization and analysis, SEM and EDS samples were coated with platinum and observed at an acceleration of 15 kV. EDS was employed for the elemental analysis of silver nanoparticle composites. The Optical microscope (OM) was primarily utilized to observe the various media layer fibers produced during different Electrospinning times. An Instron tensile tester (QC–H21B1-S00) was employed to characterize the stretchable conductors’ mechanical properties. The thermal stability of the composite ES nanofibers was analyzed using Thermogravimetric analysis (TGA, NETZSCH, Dubai, UAE). A custom-built automated two-probe measurement system connected to a Keithley 2400 Source Meter was used to measure the electrical conductivity of the stretchable conductor samples. At least ten samples were measured to calculate the average and enhance accuracy.

## 3. Results and Discussion

### 3.1. Sandwich Structure Piezoresistive Woven Nanofabric (SSPWN) Structural Analysis

#### 3.1.1. Scanning Electron Microscope (SEM) Analysis of the Sandwich Structure Piezoresistive Woven Nanofabric

The Sandwich Structure Piezoresistive Woven Nanofabric (SSPWN), as depicted in Figure 2a, the active layer with conductive fibers (SBS/rGO-AgNPs) is primarily composed in the first and third layers, sandwiching the dielectric layer (SBS) in the middle. The working principle of the SSPWN element is as follows: (1) Charge transfer is facilitated by the interwoven conductive fibers (SBS/rGO-AgNPs). (2) The elastic polymer dielectric layer (SBS) in the middle provides insulation between the first and third conductive layers due to its elastic nature, preventing charge transfer and forming a loop. The fiber-to-fiber stacking can be adjusted by varying the electrospinning time of the dielectric layer. (3) Applying pressure to the SSPWN element creates gaps between fibers in the dielectric layer (SBS) due to different stacking densities, allowing the conductive layers (first and third layers) to make contact and form a conductive path. (4) When pressure is released, the elastic property of the SBS dielectric layer causes the conductive layers to separate again, breaking the conductive path and restoring the original state, as shown in Figure 2b.

Next, we performed a detailed analysis of the conductive active layer (SBS/rGO-AgNPs), as shown in Figure 2c,d. In the images, fiber membrane structures are observed, primarily attributed to the electrospun fibers of SBS blended with reduced graphene oxide (rGO) and then subjected to in-situ reduction with silver nanoparticles (AgNPs) composite technology. These fibers may exhibit various morphologies, including individual or fiber bundles, as shown in Figure 2c. In Figure 2d, numerous silver nanoparticles can be observed dispersed on the rGO additive. The peeled-off rGO flakes also serve as a significant supporting surface for scattered Ag-ions. Therefore, adding rGO enables the in-situ reduction of Ag-ions, resulting in uniform distribution and anchoring of AgNPs on the rGO nanosheets. However, the formation of clusters and interactions between silver nanoparticles and rGO additives is evident. While aggregation (particle clusters) might typically weaken the conductivity in certain areas, from Section 3.1.2 (Figure 3), it can be observed that silver nanoparticles between the fibers are uniformly attached to the surface, forming a layer of silver film or coverage of silver nanoparticles. The aggregation of AgNPs on rGO forms a “junction” at the gaps between adjacent NPs, creating a connected network of silver nanoparticles and enhancing the conductivity. Therefore, the addition of rGO provides conductivity and promotes the formation of junctions, enhancing mechanical strength and stability, making this structure widely applicable in areas such as sensors and nanoelectronic devices.

#### 3.1.2. Conductivity Layer Surface Structure Analysis of the Electrospun Membranes (SEM and EDS Analysis)

In the analysis of the conductive layer, we utilized Scanning Electron Microscopy (SEM) and Energy Dispersive Spectrometer (EDS) for surface structure analysis of the electrospun membranes. As shown in Figure 3a, the surface morphology of the conductive layer (SBS/rGO-AgNPs) can be observed. Through the electrospinning technique, the electrospun fibers of SBS blended with rGO can be controlled to form a “non-woven fabric” structure. We observed the distribution of AgNPs, which may exhibit irregular arrangements or even distribution, mainly attributed to the dispersion of rGO in the nanofibers. Additionally, rGO particles or flake-like structures can be observed on the surface of the SBS fibers. It is well-known that the oxygen-containing functional groups on the basal planes and edges of rGO nanosheets play a crucial role in chemical reactions. These oxygen-containing groups can act as a scaffold for binding metal ions and NPs. During the preparation process, before introducing DMF, a water solution is added to the rGO water suspension, and the oxygen-containing functional groups on rGO provide active/nucleation sites to absorb positive silver ions, facilitating the in-situ reduction of AgNPs on the fiber surface.

As shown in Figure 3b, in the EDS elemental analysis, we can confirm the presence of carbon (C) and silver (Ag) elements as the main constituents in SBS/rGO-AgNPs. The carbon element comes from SBS copolymer and rGO, while the silver element indicates the presence of AgNPs and confirms the successful reduction of Ag-ions. Furthermore, through SEM mapping with Energy Dispersive Spectrometer (EDS), we can better understand the distribution of different elements in the SBS/rGO-AgNPs sample. As shown in Figure 3c, we can directly observe the uniform distribution of carbon elements in rGO, which helps us understand the distribution and content of rGO in the nanofiber membrane. At the same time, the presence of silver (Ag) elements in the sample is visible, indicating the even distribution of silver dots or particles in the sample (as shown in Figure 3d).

### 3.2. FTIR Analysis of Composite Materials

Figure 4 mainly explores the FTIR absorption spectra of SBS/rGO-AgNPs, SBS, and rGO. In the spectra, we can observe C–H stretching vibrations in the range of 2800–3000 cm^−1^, confirming the presence of carbon–hydrogen chains from styrene and butadiene moieties in both SBS/rGO-AgNPs and SBS copolymer. The appearance of C=C double bond stretching vibrations in the range of 1600–1660 cm^−1^ indicates the presence of styrene in both samples. Furthermore, the C=C double bond bending vibrations of 800–900 cm^−1^ represent the carbon-carbon double bonds in the benzene rings.

Differences between SBS/rGO-AgNPs and rGO can be identified by analyzing the FTIR absorption peaks. As a two-dimensional material, rGO’s characteristic peaks mainly include C–H stretching vibrations and C=C double bond stretching vibrations. C–H stretching vibrations typically appear in the 2800–3000 cm^−1^ range, representing the carbon–hydrogen chains surrounding the carbon atoms. C=C double bond stretching vibrations occur in the 1400–1700 cm^−1^ range, representing the conjugated double bonds between carbon atoms in SBS/rGO-AgNPs and rGO.

The prominent feature peak in the range of 1580–1600 cm^−1^ corresponds to the G band (Graphene band) of rGO, confirming the presence of rGO. Based on this information, we speculate that the reduced intensity of characteristic peaks in the range of 1580–1600 cm^−1^ and 2800–3000 cm^−1^ in SBS/rGO may be attributed to the interaction between C–H stretching vibrations of SBS and C=C double bond stretching vibrations (purple line) as well as the C–H stretching vibrations (2800–3000 cm^−1^) and G band (1580–1600 cm^−1^) of rGO. Moreover, silver nanoparticles typically do not exhibit significant absorption peaks in the FTIR spectra due to their low absorbance in the infrared range. Hence, this could be a substantial reason for the rapid reduction in peak intensity below 500 nm.

### 3.3. Thermal Stability Thermogravimetric Analysis (TGA)

In Figure 5, we tested thermal stability using TGA on pure SBS, rGO, the SBS/rGO nanofibers, and the SBS/rGO-AgNPs composite nanofibers. Corresponding to the thermal decomposition temperature of SBS, the TGA curve of pure SBS shows a significant mass loss peak at approximately 330 °C. During the thermal decomposition, SBS polymer chains break, releasing volatile substances and low-molecular-weight products, resulting in mass loss. On the other hand, rGO exhibits excellent thermal stability under the same testing conditions, showing no significant mass loss. This is due to the inherent high thermal stability and thermal conductivity of rGO. Adding rGO to SBS, an improvement in the thermal stability of the SBS/rGO composite material is observed. Specifically, as the rGO content increases to 0.7 wt%, the thermal decomposition temperature increases, and the mass loss decreases. This indicates that adding rGO enhances the thermal stability of SBS, possibly by absorbing and dispersing heat and inhibiting the thermal decomposition of polymer chains. Furthermore, we observed a rapid change in the thermal weight loss at temperatures above 400 °C after forming the rGO/SBS-AgNPs composite material. We speculate that this phenomenon is mainly attributed to silver ion precursors’ adsorption and chemical bonding on the rGO (as discussed in Section 3.1.2. EDS findings). Effective connections are formed between the metallic silver nanoparticles and rGO during the in-situ reduction process, requiring more energy to break the composite structure. As a result, the thermal weight loss temperature is higher than rGO/SBS, creating a composite material with high thermal stability.

### 3.4. Mechanical Strength Properties

In the mechanical strength characterization, we conducted strength and stretchable tests on each layer of the Sandwich Structure Piezoresistive Woven Nanofabric (SSPWN). In Table 1, we primarily tested the tensile strength at break (MPa) and elongation at break (%) for the conductive active layer (SBS/rGO-AgNPs), the dielectric layer (SBS), and the complete SSPWN structure. From the values obtained, it can be observed that the incorporation of rGO and AgNPs enhances the mechanical strength properties of the composite fibers while retaining their essential stretchability. The material also exhibits considerable toughness (as indicated by the Toughness value). The main reason for this improvement can be attributed to the introduction of rGO and AgNPs, which not only form “junctions” due to the aggregation of AgNPs on rGO but also lead to interactions between the C=C double bond stretching vibrations of SBS and the C–H stretching vibrations and G band of rGO (as confirmed in Section 3.1 and Section 3.2).

### 3.5. Electrical Characteristic Analysis of Sandwich Structure Piezoresistive Woven Nanofabric (SSPWN)

#### 3.5.1. The Different Dielectric Layer Nanofibers Densities Electrical Analysis

We assembled SBS/rGO-AgNPs in a sandwich structure, as shown in Figure 6a, with SBS/rGO-AgNPs as the first and third conductive layers and an SBS nanofiber membrane as the dielectric layer. We evaluated the conductivity of the conductive layer using different fabrication methods (shown in Appendix A) and the impact of dielectric layer nanofiber membrane density variation on SSPWN performance.

Appendix A shows that the conductive layer in this work exhibits significantly higher conductivity (Conductivity = 653 S/cm) than other methods. In pursuit of enhancing the efficiency of current conduction and facilitating smoother electron flow, we conducted a comparative analysis within Appendix A. This analysis encompassed a wide array of samples and diverse manufacturing processes. The results show that Graphene aerogels created through methods such as the Hummers process and Freeze-thaw exhibit considerably lower conductivity (=0.17 S/cm). Various silver composites were synthesized utilizing distinct techniques, including Reduction self-assembly/Freeze drying, Polyol synthesis/freeze casting thermal annealing, Modified Free radical polymerization/crosslinking, and Emulsion template synthesis. These methodologies yielded a range of materials, such as silver foam, silver nanowire aerogels, silver flakes—polyacrylamide-alginate hydrogen, and silver nanowire aerogels, among others. Notably, increasing the specific surface area of silver nanowires contributed to a significant rise in conductivity, with levels nearing 510 S/cm. Therefore, comparing the polymer electrospinning technology with a high specific surface area approach, it’s apparent that samples prepared through Electrospinning/in-situ AgNPs methods exhibit elevated conductivity. In this investigation, the incorporation of rGO imparts substantial toughness to the material (as corroborated in Section 3.4.) and preserves a high conductivity (=653 S/cm).

Figure 6b compares the current change under the same applied pressure for different dielectric layer nanofiber membrane densities achieved through different SBS electrospinning times (The inset shows the optical microscope images of varying nanofiber densities, approximately 1–6 min, respectively). We found that as the dielectric layer nanofiber membrane density increased, the SSPWN could withstand more significant pressure and release a higher current. A higher nanofiber membrane density in the dielectric layer provides a higher pressure detection limit (∆I/I_0_ = 8 × 10^−1^), enabling the SSPWN to respond more accurately to external stimuli. We observed that when the dielectric layer nanofiber membrane density was low, it had a lower pressure detection limit (∆I/I_0_ = 2 × 10^−1^), resulting in a smaller released current.

#### 3.5.2. Sensing Characteristics of a Multifunctional SSPWN under Mechanical Stimuli

We evaluated the sensing characteristics of a multifunctional SSPWN under pressure stimuli. Figure 7a depicts the variation of SSPWN in response to mechanical stimulation. When considering the “Unloading” scenario, we can conceptualize the Sensor as being in the OFF state of a switch. Following the application of Loading, the switch initiates a transition to the Normally Open (NO) state. In Figure 7b, we demonstrate the stability of SSPWN after over 5500 load testing cycles under an applied pressure of 0.17 kPa. The test results show that our device exhibits exceptionally high stability and sensitivity. Even after 5500 cycles, SSPWN maintains excellent repeatability and reproducibility under applied pressure without any delay phenomenon. Throughout the entire cyclic testing process, the variation in resistance value signal is highly stable, indicating the long-term stability and reliability of our SSPWN. Figure 7c shows the variation in resistance value during the cyclic testing. Applying 0.17 kPa pressure to SSPWN showed a resistance value change of approximately 10^2^–10^3^ ohms. Although it does not generate a more extensive response on-off range, it already demonstrates a considerable level of performance for pressure sensing and small-scale switches. Additionally, Figure 7d shows that the response time of SSPWN is less than 0.03 s, clearly demonstrating its excellent response time characteristics. This is a critical factor in tactile application research. Therefore, our SSPWN offers outstanding advantages in multiple fields with its simple and practical design.

### 3.6. SSPWN Application on Human Motion Monitoring

#### 3.6.1. Motion Monitoring of the Different Areas of the Hand

In Figure 8, we performed tests on different areas of the hand (fingers, back of the hand, wrist, and elbow). Due to the stretching, bending, and pressure exerted on the skin of the hand, we observed that the relative resistance (∆R/R_0_) increased with the increasing bending angle, corresponding to the tensile strain and bending pressure. Similarly, as the range of motion in different hand regions increased, the resistance value also increased, demonstrating the potential application of SSPWN in a wide range of human motion monitoring.

#### 3.6.2. Foot Motion Monitoring of RGB Sensing Shoes

We applied the Sandwich Structure Piezoresistive Woven Nanofabric (SSPWN) to create an innovative RGB sensing shoe sensor, as shown in Figure 9. This sensor combines excellent mechanical performance and conductivity, demonstrating outstanding functionality. During the fabrication process, we integrated the SSPWN with an RGB LED strip and arranged the LED strip on the RGB sensing shoes using a specific configuration. Finally, a battery was used as the power source to supply the required power to the LED strip. Figure 9 clearly shows the completed RGB sensing shoes, presenting the overall appearance and structure of the device. The LEDs remain unlit when the foot is not in contact with the SSPWN. However, when the footsteps are on the SSPWN and applying pressure, the resistance of the SSPWN changes, allowing the circuit to conduct, and the LED strip emits light. This design provides users with an interactive and engaging wearable product. The RGB-sensing shoes can sense changes in foot pressure and interactively display them through the lighting effects. In the future, it can be applied in various fields, such as sports monitoring, health tracking, foot pressure distribution analysis, and gait analysis. We are excited about the application of electrospinning technology and electroless silver reduction in wearable products, and we believe that this innovative RGB-sensing shoe will bring a new wearing experience to people and has the potential for wide-ranging applications.

## 4. Conclusions

In this study, we harnessed the potential of nanotechnology to elevate the design of stretchable metal-organic polymer nanocomposites tailored for wearable sensors. Our approach centered on integrating reduced graphene oxide (rGO) and in-situ generated silver nanoparticles (AgNPs) within flexible electrospun polystyrene-butadiene-polystyrene (SBS) fibers, culminating in the creation of a Sandwich Structure Piezoresistive Woven Nanofabric (SSPWN). Through comprehensive SEM, EDS, FTIR, and TGA analyses, we successfully synthesized the composite material of rGO/SBS-AgNPs. A distinctive peak within the 1580–1600 cm^−1^ range unequivocally aligned with the G band (Graphene band) of rGO, validating its presence. EDS imaging further underscored the uniform dispersion of rGO and AgNPs. Regarding electrical characterization, the conductive layer within this study exhibited a substantially augmented conductivity (Conductivity = 653 S/cm) compared to alternative methodologies. Moreover, the implementation of SSPWN fabrication amplified the mechanical robustness of the material, showcasing remarkable Toughness (2.15418 MJ·m^−3^) and noteworthy Elongation properties (Elongation at break (%) = 71.35%).

The tactile-sensitive wearable sensor, founded upon the SSPWN, demonstrated commendable attributes, including rapid response times (<3 ms) and a remarkable endurance of over 5500 cycles, attesting to its enduring stability. Therefore, we successfully applied the SSPWN to human motion monitoring, including different areas of the hand and RGB sensing shoes for foot motion monitoring. The nanocomposite’s exceptional thermal stability, resulting from effective connections between rGO and AgNPs, makes it suitable for wearable electronic applications. Our findings suggest the potential for intelligent healthcare, health monitoring, gait detection, and analysis, offering exciting prospects for future wearable electronic products. Our research provides an innovative approach to polymer materials and demonstrates a path toward developing advanced wearable electronic devices.

## Figures and Tables

**Figure 1 nanomaterials-13-02375-f001:**
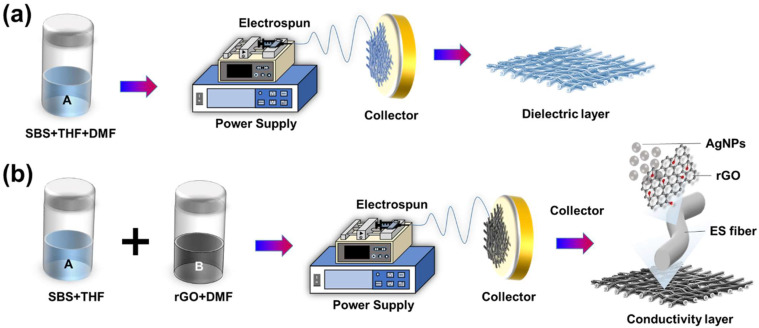
Nanofiber membrane different layer preparation process. (**a**) the dielectric layer, and (**b**) the Conductivity Layer.

**Figure 2 nanomaterials-13-02375-f002:**
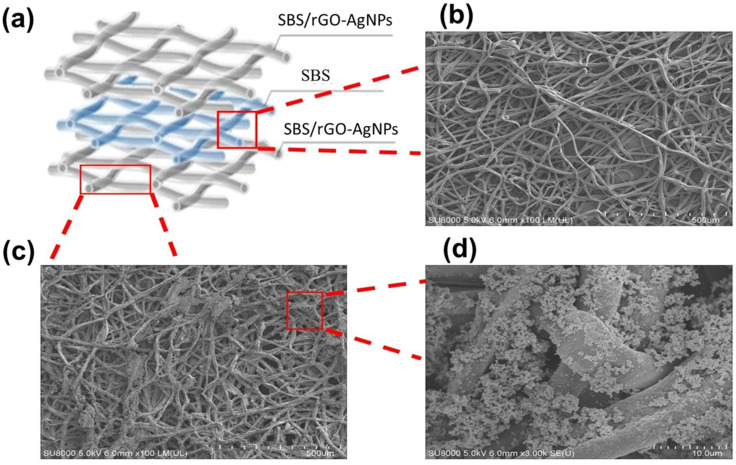
Structural analysis of Sandwich Structure Piezoresistive Woven Nanofabric (SSPWN). (**a**) Schematic diagram of the distribution of the active layer and the dielectric layer. (**b**) SEM image of the SBS dielectric layer. (**c**) The conductive active layer (SBS/rGO-AgNPs) SEM image. (**d**) Enlarged view of the conductive active layer.

**Figure 3 nanomaterials-13-02375-f003:**
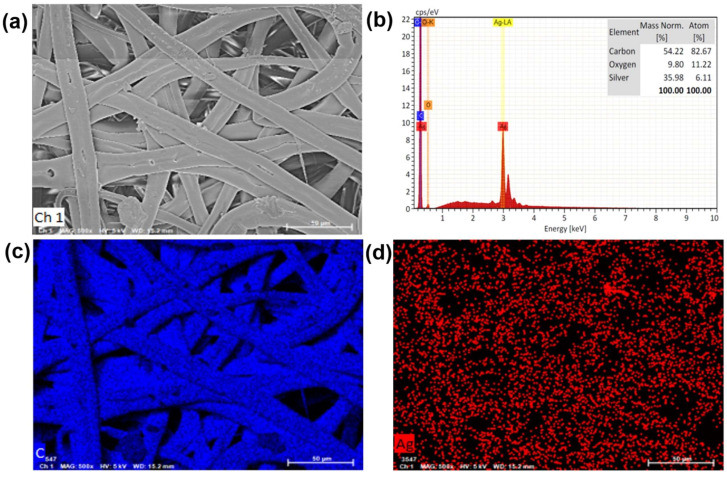
Surface elemental analysis of the conductive layer. (**a**) SEM image of the SBS/rGO-AgNPs. (**b**) EDS elemental analysis of the conductive layer. (**c**,**d**) is the mapping spectrum of the conductive layer, which shows the distribution of carbon and silver, respectively.

**Figure 4 nanomaterials-13-02375-f004:**
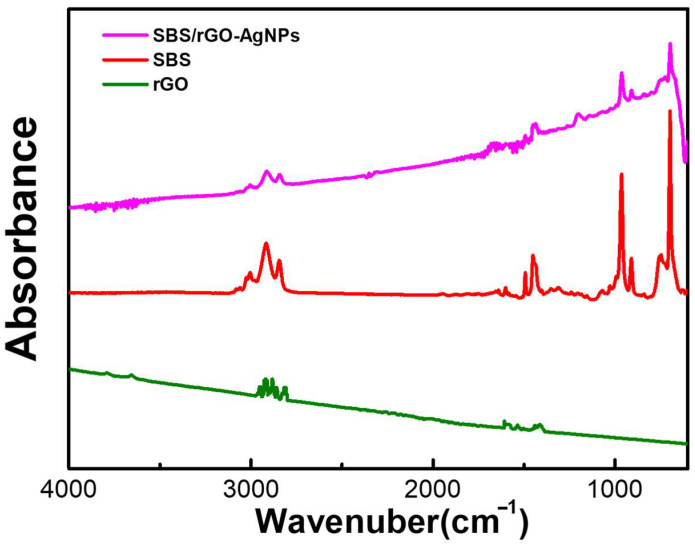
FTIR characteristics analysis of SBS/rGO-AgNPs, SBS and rGO.

**Figure 5 nanomaterials-13-02375-f005:**
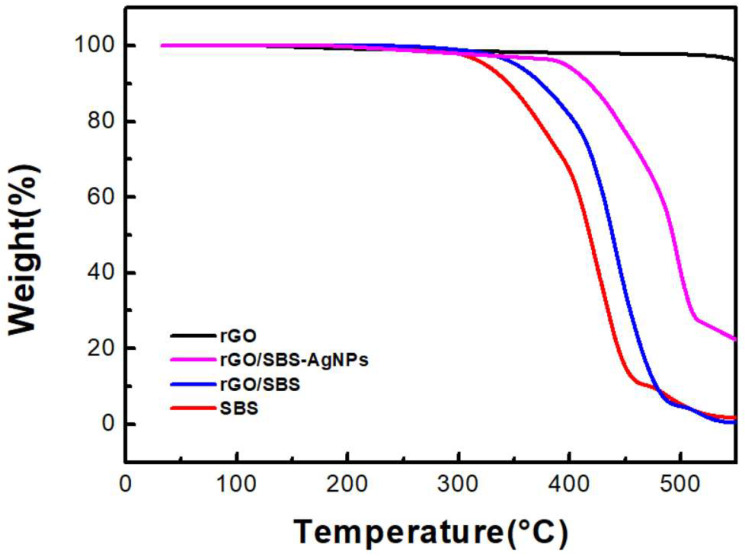
The thermal stability thermogravimetric analysis (TGA) images of pure SBS, rGO, the SBS/rGO nanofibers, and the SBS/rGO-AgNPs composite nanofibers.

**Figure 6 nanomaterials-13-02375-f006:**
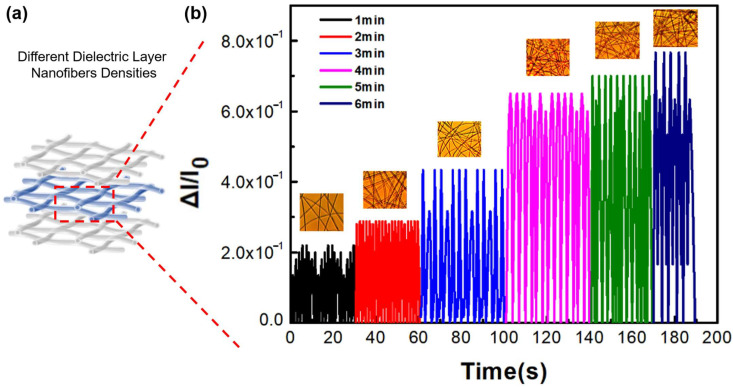
Electrical Characteristic Analysis of SSPWN. (**a**) Schematic of Sandwich Structure Piezoresistive Woven Nanofabric. (**b**) The different densities of dielectric Layer Nanofibers voltage change Analysis.

**Figure 7 nanomaterials-13-02375-f007:**
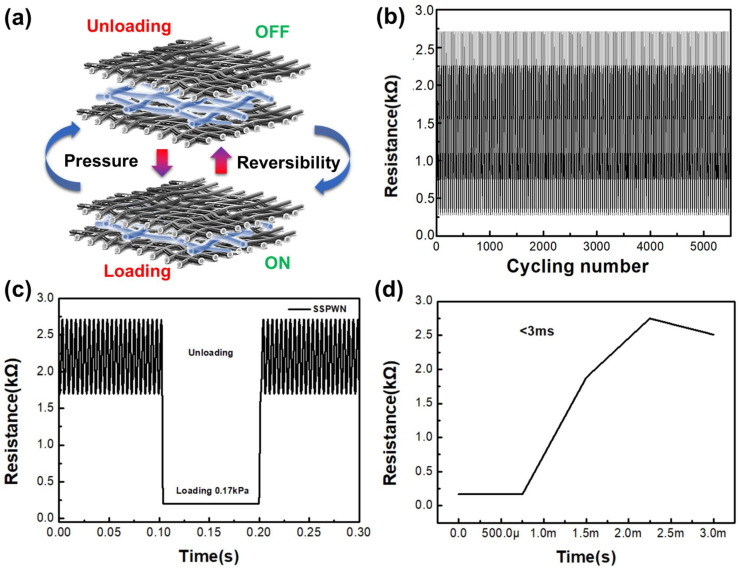
SSPWN at 0.17 kPa pressure electrical characteristic analysis. (**a**) the schematic illustrations of mechanical stimulation, (**b**) over 5500 cycles, (**c**) the On-Off response, and (**d**) the instantaneous response time.

**Figure 8 nanomaterials-13-02375-f008:**
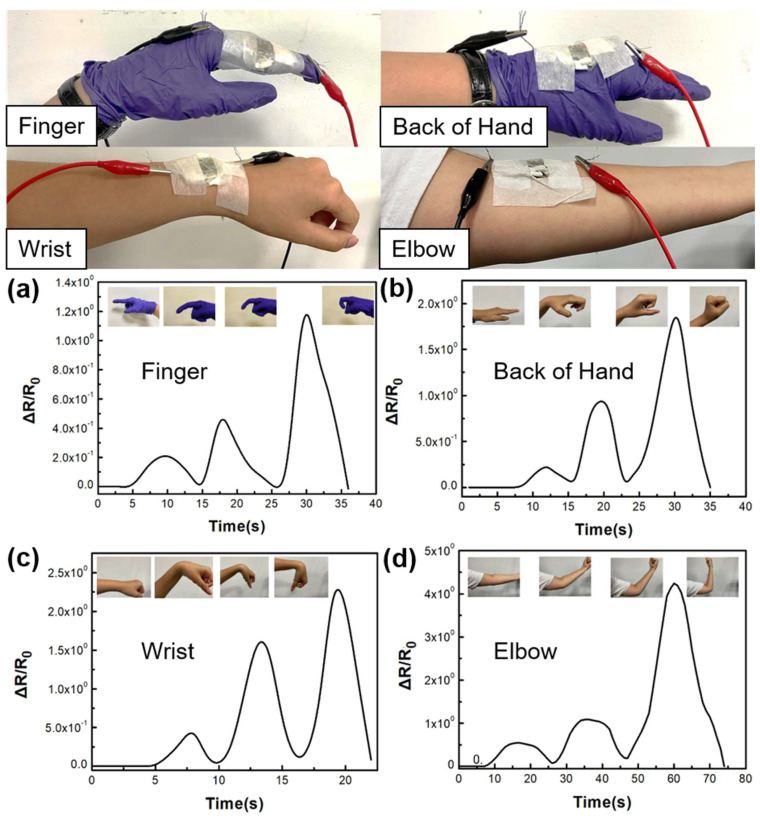
Electrical characteristic analysis relative resistance to SSPWN for (**a**) fingers, (**b**) back of the hand, (**c**) wrist, and (**d**) elbow.

**Figure 9 nanomaterials-13-02375-f009:**
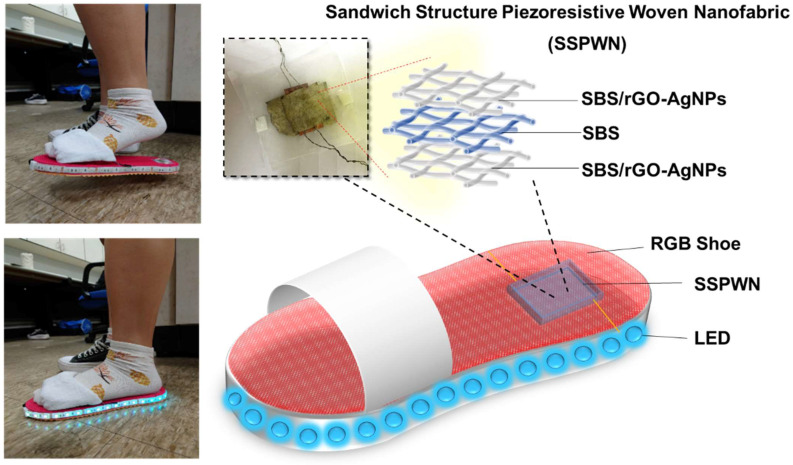
Schematic diagram of the RGB sensing shoes and the changes in LED brightness corresponding to the foot pressure applied to SSPWN.

**Table 1 nanomaterials-13-02375-t001:** Mechanical strength properties of SBS, SBS/rGO-AgNPs, and SSPWN.

Samples	Tensile Strength at Break (MPa)	Elongation at Break (%)	Toughness (MJ·m^−3^)
SBS	0.9226	140.70%	0.75092
SBS/rGO-AgNPs	2.8872	85.53%	1.43481
SSPWN	5.8701	71.35%	2.15418

## Data Availability

The data presented in this study are available on request from the corresponding author.

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
