# Peer review of "Stretchable Sensors: Novel Human Motion Monitoring Wearables"

_nanomaterials, 2023, doi:10.3390/nano13162375_

Round 1
Reviewer 1 Report
This paper describe Sandwich Structure Piezoresistive Woven Nanofabric (SSPWN) based on a stretchable metal-organic nanocomposite comprising reduced graphene oxide (rGO) and in-situ generated silver nanoparticles (AgNPs). rGO is blended with elastic electrospun polystyrene-butadiene-polystyrene (SBS) fibers in order to lead to a tactile-sensitive wearable sensor. This sensor exhibits stability in 55,000 cycles and response time less than milliseconds.
A rather exhaustive set of characterization techniques has been implemented (scanning electron microscope with EDS and optical microscope, FTIR, tensile tester, electrical conductivity and thermogravimetry). In terms of materials, the various elements as well as the final product are well described.
Many tests have been designed to evaluate sensing characteristics. Real-life tests are also presented. They are relatively convincing on the practical potential of the device produced.
In conclusion, this paper can be published without modification.
Author Response
Thanks for the reviewer's deep consideration and Comments on our manuscript.

Reviewer 2 Report
Author reported the synthesis of stretchable sensor for motion monitoring. This area of sensor for monitoring have been a topic of discussion for all sort of devices (fitness or medical). The manuscript is well written and good presentation that will interest readers. However, I found that some revisions are needed before acceptance.
1. The title of the article seems to be strong when using the word 'Pioneering'. I am sure there are many human motion monitoring sensors available in the market. Authors are suggested to use appropriate wording so that it should be reasonable to read.
2. Also, author demonstrated some motion testing of the developed sensor. However, stretchability of the sensor is missing from the article when title says, 'stretchable sensors'.
3. Author discussed about sandwich structure of piezoresistive woven nanofabric. How did author achieve sandwich structure fibers being delicate at nanoscale? How many numbers of layer can be used to achieve desired results.
4. The results of figure 9 were missing from the text.
Author Response
A point-by-point revision according to the reviewers' comments
Reviewer 2:
Comments and Suggestions for Authors:
Author reported the synthesis of stretchable sensor for motion monitoring. This area of sensor for monitoring have been a topic of discussion for all sort of devices (fitness or medical). The manuscript is well written and good presentation that will interest readers. However, I found that some revisions are needed before acceptance.
- The title of the article seems to be strong when using the word 'Pioneering'. I am sure there are many human motion monitoring sensors available in the market. Authors are suggested to use appropriate wording so that it should be reasonable to read.
Answer: Thanks for the reviewer’s suggestion. The original article's title is "Stretchable Sensors: Pioneering Human Motion Monitoring Wearables." We change the title to "Stretchable Sensors: Novel Human Motion Monitoring Wearables."
- Also, author demonstrated some motion testing of the developed sensor. However, stretchability of the sensor is missing from the article when title says, 'stretchable sensors'.
Answer: Thanks for the reviewer's comment. We have added strength and stretchable tests on each layer of the Sandwich Structure Piezoresistive Woven Nanofabric (SSPWN). This information is explained in the sections "3.4. Mechanical strength properties" and "Table 1. Elongation at break (%)"
The content is as follows:
3.4. Mechanical strength properties
In the mechanical strength characterization, we conducted strength and stretchable tests on each layer of the Sandwich Structure Piezoresistive Woven Nanofabric (SSPWN). In Table 1, we primarily tested the tensile strength at break (MPa) and elongation at break (%) for the conductive active layer (SBS/rGO-AgNPs), the dielectric layer (SBS), and the complete SSPWN structure. From the values obtained, it can be observed that the incorporation of rGO and AgNPs enhances the mechanical strength properties of the composite fibers while retaining their essential stretchability. The material also exhibits considerable toughness (as indicated by the Toughness value).
Table 1. Mechanical strength properties of SBS, SBS/rGO-AgNPs, and SSPWN.
|
Samples |
Tensile strength at break (MPa) |
Elongation at break (%) |
Toughness (MJ.m-3) |
|
SBS |
0.9226 |
140.70% |
0.75092 |
|
SBS/rGO-AgNPs |
2.8872 |
85.53% |
1.43481 |
|
SSPWN |
5.8701 |
71.35% |
2.15418 |
- Author discussed about sandwich structure of piezoresistive woven nanofabric. How did author achieve sandwich structure fibers being delicate at nanoscale? How many numbers of layer can be used to achieve desired results.
Answer: Thanks for the reviewer's deep consideration of our manuscript. We have added the nanofibers formed via the electrospinning technique for the SBS Dielectric layer and the SBS-rGO nanofiber, encompassing different time intervals. This information is explained in the sections "2.2. Preparation of Dielectric layer by Electrospun SBS Nanofiber Membrane" and "2.3. Preparation of SBS/rGO Nanofiber Membrane by Electrospinning."
In addition, we can achieve desired results just need three layers. We assembled SBS/rGO-AgNPs in a sandwich structure, with SBS/rGO-AgNPs as the first and third conductive layers and an SBS nanofiber membrane as the dielectric layer. We added this information and explained it in the sections "3.5.1. The Different Dielectric Layer Nanofibers Densities Electrical Analysis"
The content is as follows:
2.2. Preparation of Dielectric layer by Electrospun SBS Nanofiber Membrane.
SBS was dissolved in a mixture of THF and DMF (at a ratio of 3:1) to prepare a 15 wt% SBS solution. The solution was then placed on a heating plate at 160 rpm and 60 degrees Celsius for 8 hours. Subsequently, the electrospinning technique was employed to create SBS nanofibrous dielectric layers. The dissolved SBS solution was injected into a metal needle using an infusion pump (KD Scientific Model 100, USA) at 0.5 to 0.8 milliliters per minute. The tip of the metal needle was connected to a high-voltage power supply (Chargemaster CH30P SIMCO, USA), and the voltage during the electrospinning process was set at 13.0 to 15.0 kV. Finally, an aluminum foil was placed 15 centimeters below the needle tip to collect the SBS nanofibrous dielectric layer produced at different time intervals (1, 2, 3, 4, 5, and 6 minutes, respectively) as shown in Figure 1(a).
2.3. Preparation of SBS/rGO Nanofiber Membrane by Electrospinning
Firstly, (1) in the preparation of the solution, SBS and THF were dissolved together to form solution A. Simultaneously, we mixed reduced Graphene Oxide (rGO) dispersed in DMF (weight percentage of 0.7 wt%) to form solution B. Subsequently, solution A and solution B were mixed after undergoing ultrasonic vibration, resulting in a mixed solution containing 15 wt% SBS (dissolved and dispersed in a solvent mixture of THF: DMF at a ratio of 3:1). The mixed solution was vigorously stirred and then placed on a heating plate at 60 degrees Celsius and 160 rpm for 8 hours to ensure complete dissolution and homogenization.
(2) Electrospinning technique was employed to fabricate SBS-rGO nanofiber thin films: The homogenized SBS-rGO solution was injected into a metal needle using an infusion pump (KD Scientific Model 100, USA) at a rate of 0.35 to 0.5 milliliters per minute. The tip of the metal needle was connected to a high-voltage power supply (Chargemaster CH30P SIMCO, USA), and the voltage during the electrospinning process was set at 13.0 to 15.0 kV. Finally, an aluminum foil was placed 15 centimeters below the needle tip to collect the densely formed SBS-rGO nanofiber thin film for 30 minutes, resulting in the SBS/rGO Nanofiber membrane.
3.5.1. The Different Dielectric Layer Nanofibers Densities Electrical Analysis
We assembled SBS/rGO-AgNPs in a sandwich structure, as shown in Figure 6(a), with SBS/rGO-AgNPs as the first and third conductive layers and an SBS nanofiber membrane as the dielectric layer. We assessed the impact of dielectric layer nanofiber membrane density variation on SSPWN performance. Figure 6(b) compares the cur-rent change under the same applied pressure for different dielectric layer nanofiber membrane densities achieved through different SBS electrospinning times (The inset shows the optical microscope images of varying nanofiber densities). We found that as the dielectric layer nanofiber membrane density increased, the SSPWN could with-stand more significant pressure and release a higher current. A higher nanofiber mem-brane density in the dielectric layer provides a higher pressure detection limit, ena-bling the SSPWN to respond more accurately to external stimuli. We observed that when the dielectric layer nanofiber membrane density was low, it had a lower pressure detection limit, resulting in a smaller released current.
- The results of figure 9 were missing from the text.
Answer: Thanks for the reviewer comment. We have added the results of Figure 9 in "3.6.2. Foot Motion Monitoring of RGB sensing shoes."
The content is as follows:
3.6.2. Foot Motion Monitoring of RGB sensing shoes.
We applied the Sandwich Structure Piezoresistive Woven Nanofabric (SSPWN) to create an innovative RGB sensing shoe sensor, as shown in Figure 9. This sensor combines excellent mechanical performance and conductivity, demonstrating outstanding functionality. During the fabrication process, we integrated the SSPWN with an RGB LED strip and arranged the LED strip on the RGB sensing shoes using a specific configuration. Finally, a battery was used as the power source to supply the required power to the LED strip. Figure 9 clearly shows the completed RGB sensing shoes, presenting the overall appearance and structure of the device. The LEDs remain unlit when the foot is not in contact with the SSPWN. However, when the footsteps are on the SSPWN and applying pressure, the resistance of the SSPWN changes, allowing the circuit to conduct, and the LED strip emits light. This design provides users with an interactive and engaging wearable product. The RGB-sensing shoes can sense changes in foot pressure and interactively display them through the lighting effects. In the future, it can be applied in various fields, such as sports monitoring, health tracking, foot pressure distribution analysis, and gait analysis. We are excited about the application of electrospinning technology and electroless silver reduction in wearable products, and we believe that this innovative RGB-sensing shoe will bring a new wearing experience to people and has the potential for wide-ranging applications.

Reviewer 3 Report
1. It is necessary to correct and supplement the annotation with more specific data that reflect the essence of the question posed in the study.
2. The review does not pay attention to the use of carbon nanotubes in stretchable polymer composites: Shchegolkov A.V., Shchegolkov A.V., Zemtsova N.V. Investigation of heat release in nano modified elastomers during stretching and torsion under the action of electric voltage. Frontier Materials & Technologies. 2022;(2):121-132. (((In Rus.) https://doi.org/10.18323/2782-4039-2022-2-121-132
3. Add a comparison table with the data of other researchers.
4. To concretize conclusions and give more numerical information.
Moderate editing of English language required
Author Response
A point-by-point revision according to the reviewers' comments
Reviewer 3:
Comments and Suggestions for Authors:
- It is necessary to correct and supplement the annotation with more specific data that reflect the essence of the question posed in the study.
Answer: Thanks for the reviewer’s comment. We have enriched the specific data in Section 3.5.1, focusing on the Electrical Analysis of the Conductive layer and Different Dielectric Layer Nanofiber Densities. Additionally, in Section 3.5.2, we have expanded our discussion regarding the Sensing Characteristics of the Multifunctional SSPWN Under Mechanical Stimuli, specifically delving into the variation of SSPWN in response to mechanical stimulation.
The content is as follows:
3.5.1. The Different Dielectric Layer Nanofibers Densities Electrical Analysis
We assembled SBS/rGO-AgNPs in a sandwich structure, as shown in Figure 6(a), with SBS/rGO-AgNPs as the first and third conductive layers and an SBS nanofiber membrane as the dielectric layer. We evaluated the conductivity of the conductive layer using different fabrication methods (shown in Table S1.) and the impact of dielectric layer nanofiber membrane density variation on SSPWN performance.
Table S1 shows that the conductive layer in this work exhibits significantly higher conductivity (Conductivity = 653 S/cm) than other methods. In pursuit of enhancing the efficiency of current conduction and facilitating smoother electron flow, we conducted a comparative analysis within Table S1. This analysis encompassed a wide array of samples and diverse manufacturing processes. The results show that Graphene aerogels created through methods such as the Hummers process and Freeze-thaw exhibit considerably lower conductivity (= 0.17 S/cm). Various silver composites were synthesized utilizing distinct techniques, including Reduction self-assembly/Freeze drying, Polyol synthesis/freeze casting thermal annealing, Modified Free radical polymerization/crosslinking, and Emulsion template synthesis. These methodologies yielded a range of materials, such as Silver foam, Silver nanowire aerogels, Silver flakes - polyacrylamide-alginate hydrogen, and Silver nanowire aerogels, among others. Notably, increasing the specific surface area of Silver nanowires contributed to a significant rise in conductivity, with levels nearing 510 S/cm. Therefore comparing the polymer electrospinning technology with a high specific surface area approach, it's apparent that samples prepared through Electrospinning/in-situ AgNPs methods exhibit elevated conductivity. In this investigation, the incorporation of rGO imparts substantial toughness to the material (as corroborated in section 3.4.) and preserves a high conductivity (= 653 S/cm).
Figure 6(b) compares the current change under the same applied pressure for different dielectric layer nanofiber membrane densities achieved through different SBS electrospinning times (The inset shows the optical microscope images of varying nanofiber densities, 1min ~ 6min respectively). We found that as the dielectric layer nanofiber membrane density increased, the SSPWN could withstand more significant pressure and release a higher current. A higher nanofiber membrane density in the dielectric layer provides a higher pressure detection limit (∆I/I0 = 8 10-1), enabling the SSPWN to respond more accurately to external stimuli. We observed that when the dielectric layer nanofiber membrane density was low, it had a lower pressure detection limit (∆I/I0 = 2 10-1), resulting in a smaller released current.
3.5.2. Sensing Characteristics of a Multifunctional SSPWN Under Mechanical Stimuli
We evaluated the sensing characteristics of a multifunctional SSPWN under pressure stimuli. Figure 7(a) depicts the variation of SSPWN in response to mechanical stimulation. When considering the "Unloading" scenario, we can conceptualize the Sensor as being in the OFF state of a switch. Following the application of Loading, the switch initiates a transition to the Normally Open (NO) state. In Figure 7(b), we demonstrate the stability of SSPWN after over 5,500 load testing cycles under an applied pressure of 0.17 kPa. The test results show that our device exhibits exceptionally high stability and sensitivity. Even after 5,500 cycles, SSPWN maintains excellent repeatability and reproducibility under applied pressure without any delay phenomenon. Throughout the entire cyclic testing process, the variation in resistance value signal is highly stable, indicating the long-term stability and reliability of our SSPWN. Figure 7(c) shows the variation in resistance value during the cyclic testing. Applying 0.17 kPa pressure to SSPWN showed a resistance value change of approximately 102-103 ohms. Although it does not generate a more extensive response on-off range, it already demonstrates a considerable level of performance for pressure sensing and small-scale switches. Additionally, Figure 7(d) shows that the response time of SSPWN is less than 0.03 seconds, clearly demonstrating its excellent response time characteristics. This is a critical factor in tactile application research. Therefore, our SSPWN offers outstanding advantages in multiple fields with its simple and practical design.
- The review does not pay attention to the use of carbon nanotubes in stretchable polymer composites: Shchegolkov A.V., Shchegolkov A.V., Zemtsova N.V. Investigation of heat release in nano modified elastomers during stretching and torsion under the action of electric voltage. Frontier Materials & Technologies. 2022;(2):121-132. (((In Rus.) https://doi.org/10.18323/2782-4039-2022-2-121-132
Answer: Thank you for the valuable suggestions provided by the reviewer. We have incorporated pertinent information into the manuscript's introduction. Additionally, we also added the info in References 1. This augmentation has enriched the manuscript's depth and scope. Your proactive guidance has contributed significantly to enhancing our work's overall quality and comprehensiveness. We are grateful for your support and are confident that these enhancements will result in a more robust and informative manuscript.
The content is as follows:
- Introduction
The fields of intelligent healthcare and health monitoring have become prominent research areas. Medical monitoring systems require vital characteristics such as mechanical flexibility, stretchability, high sensitivity, and rapid response [1-5]. These features have garnered significant attention in disease diagnosis, elderly care, health monitoring, and intelligent robotics [6-8].
References
- Shchegolkov, A.V.; Zemtsova, N.V. Investigation of heat release in nanomodified elastomers during stretching and torsion under the action of electric voltage. Mater. Tech. 2022, 2, 121-132. doi:10.18323/2782-4039-2022-2-121-132.
- Add a comparison table with the data of other researchers.
Answer: Thanks for the reviewer’s suggestion. We have included a comparative table featuring other researchers' data in "Supporting Information Table S1." Furthermore, we have engaged in a comprehensive discussion of the pertinent comparative data, incorporating it into section 3.5.1 of the manuscript. Our primary focus throughout this endeavor has been assessing conductivity variations within the conductive layer, employing diverse fabrication methods.
The content is as follows:
3.5.1. The Different Dielectric Layer Nanofibers Densities Electrical Analysis
We assembled SBS/rGO-AgNPs in a sandwich structure, as shown in Figure 6(a), with SBS/rGO-AgNPs as the first and third conductive layers and an SBS nanofiber membrane as the dielectric layer. We evaluated the conductivity of the conductive layer using different fabrication methods (shown in Table S1.) and the impact of dielectric layer nanofiber membrane density variation on SSPWN performance.
Table S1 shows that the conductive layer in this work exhibits significantly higher conductivity (Conductivity = 653 S/cm) than other methods. In pursuit of enhancing the efficiency of current conduction and facilitating smoother electron flow, we conducted a comparative analysis within Table S1. This analysis encompassed a wide array of samples and diverse manufacturing processes. The results show that Graphene aerogels created through methods such as the Hummers process and Freeze-thaw exhibit considerably lower conductivity (= 0.17 S/cm). Various silver composites were synthesized utilizing distinct techniques, including Reduction self-assembly/Freeze drying, Polyol synthesis/freeze casting thermal annealing, Modified Free radical polymerization/crosslinking, and Emulsion template synthesis. These methodologies yielded a range of materials, such as Silver foam, Silver nanowire aerogels, Silver flakes - polyacrylamide-alginate hydrogen, and Silver nanowire aerogels, among others. Notably, increasing the specific surface area of Silver nanowires contributed to a significant rise in conductivity, with levels nearing 510 S/cm. Therefore comparing the polymer electrospinning technology with a high specific surface area approach, it's apparent that samples prepared through Electrospinning/in-situ AgNPs methods exhibit elevated conductivity. In this investigation, the incorporation of rGO imparts substantial toughness to the material (as corroborated in section 3.4.) and preserves a high conductivity (= 653 S/cm).
Table S1. The conductivity of the conductive layer using different fabrication methods.
|
NO. |
Samples |
Methodology |
Conductivity ( S/cm ) |
Ref. |
|
1 |
Graphene aerogelse |
Hummers method/ Freeze thawe |
0.17 |
[1] |
|
2 |
Silver foam |
Reduction self-assembly/Freeze dryingt |
170 |
[2] |
|
3 |
Silver nanowire aerogels |
Polyol synthesis/ freeze casting thermal annealing |
510 |
[3] |
|
4 |
Silver flakes- polyacrylamide- alginate hydroge |
Modified Free radical polymerization/ crosslinking |
374 |
[4] |
|
5 |
Silver nanowire aerogels |
Emulsion template synthesist |
66 |
[5] |
|
6 |
polyurethane/PPy (PU/PPy) |
Electrospinning/in-situ chemical polymerization |
276 |
[6] |
|
7 |
PEDOT |
electrospinning/vapor-phase polymerization |
60 |
[7] |
|
8 |
Silver nanowires in polyvinyl alcohol |
Electrospinning/ in-situ AgNPs |
650 |
[8] |
|
9 |
SBS/rGO-AgNPs |
Electrospinning/in-situ AgNPs |
653 |
This work |
- To concretize conclusions and give more numerical information.
Answer: Thanks for the reviewer's valuable comment. We have taken steps to concretize our conclusions and provide more detailed numerical information, ensuring the robustness and clarity of our findings.
The content is as follows:
- Conclusion
In this study, we harnessed the potential of nanotechnology to elevate the design of stretchable metal-organic polymer nanocomposites tailored for wearable sensors. Our approach centered on integrating reduced graphene oxide (rGO) and in-situ generated silver nanoparticles (AgNPs) within flexible electrospun polystyrene-butadiene-polystyrene (SBS) fibers, culminating in the creation of a Sandwich Structure Piezoresistive Woven Nanofabric (SSPWN). Through comprehensive SEM, EDS, FTIR, and TGA analyses, we successfully synthesized the composite material of rGO/SBS-AgNPs. A distinctive peak within the 1580-1600 cm-1 range unequivocally aligned with the G band (Graphene band) of rGO, validating its presence. EDS imaging further underscored the uniform dispersion of rGO and AgNPs. Regarding electrical characterization, the conductive layer within this study exhibited a substantially augmented conductivity (Conductivity = 653 S/cm) compared to alternative methodologies. Moreover, the implementation of SSPWN fabrication amplified the mechanical robustness of the material, showcasing remarkable Toughness (2.15418 MJ.m-3) and noteworthy Elongation properties (Elongation at break (%) = 71.35%).
The tactile-sensitive wearable sensor, founded upon the SSPWN, demonstrated commendable attributes, including rapid response times (< 3 ms) and a remarkable endurance of over 5,500 cycles, attesting to its enduring stability. Therefore, we successfully applied the SSPWN to human motion monitoring, including different areas of the hand and RGB sensing shoes for foot motion monitoring. The nanocomposite's exceptional thermal stability, resulting from effective connections between rGO and AgNPs, makes it suitable for wearable electronic applications. Our findings suggest the potential for intelligent healthcare, health monitoring, gait detection, and analysis, offering exciting prospects for future wearable electronic products. Our research provides an innovative approach to polymer materials and demonstrates a path toward developing advanced wearable electronic devices.

Round 2
Reviewer 2 Report
The manuscript can be accepted with necessary changes being made as per the suggestions.
Reviewer 3 Report
Everything is fixed.